# The association between depression and addictive social media use during the COVID-19 pandemic: The mediating role of sense of control

**Zahir Vally**[1]* , **Mai Helmy**[2,3], **Louis Fourie**[1]

**1** Department of Clinical Psychology, United Arab Emirates University, Al Ain, United Arab Emirates,
**2** Department of Psychology, Sultan Qaboos University, Muscat, Sultanate of Oman, **3** Department of Psychology, Menoufia University, Shebin El-Kom, Egypt

* zahir.vally@uaeu.ac.ae

**Data Availability Statement:** All relevant data are within the paper and its Supporting Information files.

## Abstract

### Background

COVID-19 precipitated a plethora of mental health difficulties, particularly for those with pre-existing mental health concerns such as depression or addictive tendencies. For some, the distress that emanated from the experience of the pandemic prompted excessive engagement in the safety of online interactions on social media. The present study examined whether variation in individuals' sense of control explained the association between depression and addictive social media use.

### Method

A sample of 1322 participants from two Middle Eastern nations provided data collected during the peak of the pandemic from February to May 2021. A combination of convenience and snowball sampling were used to recruit and collect data from college-aged students enrolled at two universities in Egypt and the United Arab Emirates, respectively. This study adopted a cross-sectional design in which participants completed a self-administered survey that consisted of measures that assessed depressive affect, sense of control, and addictive social media use.

### Results

Depression was significantly and positively associated with addictive SMU. Sense of control was negatively related to both depression and SMU and significantly mediated the association between these two variables ($\beta$ = .62, SE = .03, 95%CI .56, .68).

### Conclusion

This study identified a potential protective variable that could be targeted by psychological treatment to ameliorate the potential onset of addictive SMU in individuals with depressive

**Funding:** The author(s) received no specific funding for this work.

**Competing interests:** The authors have declared that no competing interests exist.

symptoms under conditions of immense psychological distress such as a worldwide pandemic.

## Introduction

The coronavirus (COVID-19) pandemic exacted substantial changes to the daily lives of all individuals [1]. Government-instituted measures designed to curtail the spread of the disease included mandatory curfews, stay-at-home orders, the wearing of face masks, travel blockades, the closure of schools, businesses, and workplaces, and a general suspension of public life [2, 3]. Despite life now having begun to slowly return to a sense of normalcy, a plethora of research has demonstrated that the COVID-19 pandemic resulted in substantial detrimental effects on individuals' mental health, the long-term effects of which remain unclear at present [4–6]. Moreover, if future pandemics were to arise, an understanding of the factors that might serve a protective function against the onset, and potentially mitigate the exacerbation, of unfavourable mental health outcomes would be particularly beneficial to both researchers and clinicians.

### Addictive social media use

A commonly reported consequence of the requirement to engage in 'spatial distancing' was the precipitous rise in the excessive use of technology and social media platforms. As individuals were unable to engage with others in real life, most, if not all, interactions were shifted to the online realm, leading to a notable increase in the use of social media platforms in comparison to pre-pandemic rates [7, 8]. Addictive social media use (SMU), whilst not presently being a recognised diagnostic construct, reportedly results in considerable symptomology and social dysfunction [9]. It is characterised by a close emotional bond to the device and/or the platform, individuals experience an immense need to stay connected, and may experience withdrawal symptoms when not connected or when prevented from using the device/ platform [10]. Griffiths [11] contends that addictive SMU consists of the following core elements: salience (ruminative and persistent thoughts about using social media), tolerance (progressively increasing periods of usage time is required to garner similar affective experiences from the platform's use), mood modification (using social media precipitates positive emotions and/ or combats negative emotions), relapse (unsuccessful attempts to decrease use of the platform), withdrawal symptoms (feelings of discomfort when not using social media), and conflicts (the individual's SMU results in interpersonal difficulties with others).

A substantial body of cross-sectional and longitudinal evidence attests to the contention that excessive and addictive SMU is associated with a range of unfavourable physical and mental health difficulties [12, 13]. Despite the contentions of some that caution should be exercised to avoid pathologizing typical behaviours that are characteristic of contemporary society and consequently overstate its prevalence [14–16], the evidence in support of the potential deleterious consequences of excessive SMU is overwhelming. Psychological distress, depression, anxiety, and insomnia have consistently been shown to be associated with addictive SMU when assessed cross-sectionally and this is evident across Asia, Europe, and North America [6, 10, 17–19].

Additionally, when assessed longitudinally, baseline levels of addictive SMU appear predictive of psychopathological constructs (i.e., depression, insomnia, suicidal ideation) at ensuing follow-up assessment points. For example, in Germany, longitudinal research has shown

baseline levels of additive SMU to be positively associated with the presence of depressive symptoms at six weeks follow-up [20]. Two further studies have demonstrated that addictive SMU at baseline was associated with suicidal behaviour twelve months later [21, 22]. This is evident both in samples of individuals with pre-existing mental health difficulties [23] as well as college-aged participants drawn from a community sample [18].

## Sense of control and positive mental health

The presence of positive mental health mitigates the potentially detrimental effect of negative life experiences and acts as a protective mechanism reducing risk for the development of mental health difficulties [24–27]. This mitigating effect may be the result of the elevated sense of control and greater degree of resilience that appears to accompany the presence of positive mental health [25, 28]. Seligman [29] proposes that an elevated level of sense of control is essential for the development of positive mental health. Conversely, where individuals exhibit diminished sense of control, amplified helplessness is likely, and individuals will espouse a desire to regain the perception of control [30]. Such individuals tend to employ the use of dysfunctional coping strategies which may invariably further compound their mental health difficulties. This has been shown to be the case for individuals with substance abuse difficulties and those who engage in excessive technology use such as gaming [15, 31].

There is further evidence that some individuals who experience a low sense of control may resort to seeking control in alternate contexts such as via online activities and interactions [32]. This behaviour is often reinforced, and may become more intensive and excessive, as it tends to produce positive emotions and commensurately reduces the experience of negative emotions, at least in the short-term (e.g., diminished loneliness, depression, anxiety and greater self-reported life satisfaction and wellbeing) [7, 10, 33, 34]. However, in the long-term, the individuals' excessive engagement in online interactions is likely to progress and contribute to the development addictive tendencies. The Interaction of Person-Affect-Cognition-Execution (I-PACE) model of addictive behaviour contends that a multitude of psychological and neurobiological factors collectively and cumulatively contribute to the onset or mitigation of addictive tendencies [35]. Thus, it is imperative that studies investigate the range of factors that might moderate and/ or mediate this association and their potential interaction.

It is likely that unhealthy manifestations of mental health such as depression or, indeed, factors that are beneficial for mental health fall within the parameters of the I-PACE model. This contention is supported by the following evidence. A wealth of evidence has shown that depression is positively related to addictive SMU [17, 20, 23]. Additionally, sense of control has also been demonstrated to be related to addictive SMU [28, 32, 36]. Moreover, Seligman [29] proposed that elevated levels of sense of control can have an analgesic effect on mental health outcomes. In other words, sense of control mitigates poor mental health and contributes to positive mental health. This is a contention for which there is much substantiating evidence [37–40].

Research conducted during the COVID-19 pandemic also provides further substantiating evidence of this relationship. Where individuals' sense of control was assessed, this was shown to be positively associated with baseline levels of positive mental health assessed before the onset of COVID-19 [28]. Additionally, where individuals reported the perception of a low level of control in relation to their experience of the pandemic, they were also shown to exhibit elevated risk for addictive tendencies towards technology [36]. Considering the demonstrated associations between these variables–mental health difficulties, sense of control, and addictive SMU–and the proposition of the I-PACE model, it is likely that variations in individuals' sense of control during the midst of the pandemic would mediate the association between depressive symptoms and the onset of addictive SMU.

## Theoretical framework

The theoretical model that provides the most comprehensive and appropriate conceptualization of excessive SMU is the I-PACE model [35]. The model proposes that a number of categories of risk and prognostic factors, which the model separates into various categories, collectively impact the onset of individuals' excessive engagement with devices and/or technologies. First, the *personal determinants* category comprises factors related to genetics, biology, personality characteristics, psychopathological variables, and the motives for engaging in excessive use. Second, the *risk and resilience* category, which Brand et al. [35] describe as factors that are representative of the individual's response to the personal determinants factors, comprises cognitive and attention biases, coping strategies, expectancies, craving, and variations in inhibitory control.

The model further contends that the risk and resilience factors likely operate as mediators and moderators in the relationship between the personal determinants factors and the consequent onset of excessive SMU. They may either serve to amplify the effect of the personal determinants factors, and thus increase the likelihood of excessive SMU occurring, or they may serve a protective function and attenuate the resultant effect [35]. Given this model's propositions and the evidence of the demonstrated associations between the study's principal variables, sense of control would be regarded as a resilience factor and thus, if the model's contentions are valid, would likely serve to diminish the effect of depression on the onset of addictive SMU.

## Aims and hypotheses

This study aimed to examine the association between depressive symptomology and addictive SMU during the COVID-19 pandemic and, moreover, whether this proposed association would be mediated by sense of control. The following hypotheses were proposed. It was hypothesized that depression would be positively associated with addictive SMU Hypothesis 1a (H1a). It was also predicted that sense of control would be negatively associated with both depression (H1b) as well as addictive SMU (H1c), respectively. Finally, it was predicted that sense of control would mediate the association between depression and addictive SMU (H2).

# Materials and method

## Study design

This study employed a cross-sectional design in which participants, who agreed to participate, completed a survey. This study's conduct was approved by the Social Sciences Research Ethics Committee at the United Arab Emirates University (Reference number: ERS_2020_6102).

## Procedure and participants

This study employed a combination of convenience and snowball sampling approaches to collect data from college-aged participants during the Spring semester of the 2020/2021 academic year with participants drawn from enrolled students at two large federal universities–Menoufia University in Egypt and the United Arab Emirates University in the UAE. Data collection occurred at the peak of the pandemic in both these locations. At that time, in the UAE, strict lockdown measures were in place, universities, schools, and non-essential workplaces were closed with work-from-home being commonplace. Conversely, in Egypt, a less stringent approach was common. Wearing masks in public was not strictly enforced and schools and universities remained operating face-to-face.

The potential sampling frame comprised approximately 2000 participants, these were students who were enrolled in the classes taught by the two principal investigators (ZV and MH) across the two campuses (the targeted classes included students from clinical psychology, abnormal psychology, cognitive psychology, research skills, and creative thinking skills). Students who met the inclusion criteria (i.e., aged at least 18 years old and self-reported as a current user of at least one social media platform) were invited to participate by completing the electronic survey. Advertisements about the study were also placed on physical notice boards and posted to social media accounts typically used by this group of students. The students in the targeted classes were also encouraged to circulate the electronic link to the study's information and the survey in their social groups, thus introducing a snowball approach to the collection of data.

The link that was made available to participants enabled completion of the electronic survey. The first page displayed an informed consent form which provided background information about the study and principal investigators' contact details. It also outlined the rights of the participants and the responsibilities of the research team (e.g., that participation was voluntary, issues of confidentiality, the right to withdraw without penalty, and measures employed to secure participants' data). Written informed consent was obtained before proceeding to commencement of the survey. Data collection occurred from February to May 2021.

## Assessment instruments

**Demographics.** Participants self-reported their age, gender, relationship status, and registration status (i.e., fulltime, or part-time).

**Depression.** The depression subscale of the Depression Anxiety and Stress Scales 21 (DASS-21) [41] was used to measure depressive affect over the preceding 7 days. This subscale consists of 7 items to which participants respond using a 4-point Likert scale (0 = did not apply to me at all, 3 = applies to me very much or most of the time). Example items include "I felt that I had nothing to look forward to" and "I felt down-hearted and blue". Higher total scores are indicative of greater levels of depressive affect. The DASS-21 is one of the most prevalently employed measures of depressive and anxious affect and has been shown to be psychometrically valid and reliable across a wide range of languages and cultures including with Arabic-speaking participants [10]. This Arabic-language version typically demonstrates internal consistency values that range from .88 to .93 [42–44]. In the present study, the Cronbach's α value was similarly high (.81).

**Sense of control.** The 2-item scale first developed and employed by Brailovskaia and Margraf [37] was used to assess sense of control. The two items are: "Do you experience important areas of your life (i.e., work, free-time, family, etc.) to be uncontrollable, meaning that you cannot, or barely can, influence them?" and "Do you experience these important areas of your life as unpredictable or inscrutable?". Responses to the items are scored using a 5-point Likert scale and higher total scores are indicative of higher sense of control. The measure's Cronbach's α scores have ranged from .79 to .91 [32, 36, 45]. In the present study, internal consistency was .82.

**Addictive social media use.** Addictive SMU was measured using the 6-item Bergen Social Media Addiction Scale (BSMAS) [9]. The items of the scale measure the 6 principal component features of addiction proposed by Griffiths [11], namely, salience, tolerance, mood modification, relapse, withdrawal, and conflict. Responses to the items are scored using a 5-point Likert scale (1 = very rarely, 5 = very often). Possible total scores can range from 6 to 30 and higher total scores are indicative of more substantial addictive tendencies to SMU. Example items include "I feel an urge to use social media more and more" and I spend a lot of time

thinking about social media or planning how to use it". The scale has a demonstrated unidimensional factor structure and meets a variety of indices indicative of reliability and validity [46, 47]. In the present study, internal consistency was equally high ($\alpha$ = .80).

**Data analysis.** Descriptive results with regard to the demographic characteristics and the principal variables are reported using means and standard deviations for continuous variables or counts and percentages for categorical variables. As a preliminary investigation of the potential relationships between depression, sense of control, and addictive SMU, a correlational matrix was computed, the results of which are reported using Pearson's r values and their corresponding significance values. A mediation model was then analysed in which depression was specified as the predictor variable, sense of control as the hypothesized mediator, and addictive SMU as the outcome variable. Gender was specified as a covariate as evidence suggests gendered differences in the use of social media platforms is common.

In the proposed mediational model, path a indicates the relationship between depression and sense of control and path b represents the association between sense of control and addictive SMU. The indirect effect (ab) is reflective of the combined effect of both paths a and b. The association between depression and addictive SMU, the total effect, is represented as path c, whilst the relationship between these two variables (the predictor and outcome) following inclusion of the proposed mediator (sense of control), is indicated as path c' (the direct effect). All analyses were conducted using SPSS Version 26 and the mediation analyses were conducted using the PROCESS macro version 3.5 (www.processmacro.org/index.html) [48]. The results of all analyses were regarded as statistically significant with a p value of .05.

# Results

## Descriptive and correlational results

The final sample consisted of 1322 participants whose age ranged from 18 to 32 years (M = 19.50 years, SD = 1.54). The vast majority of the sample were fulltime students (96.4%) while 2.6% were part-time students also simultaneously engaged in minimal employment and 1.0% of the sample were recent graduates from the university but still unemployed at the time of data collection. The sample was primarily comprised of females (75.4%) and single individuals (90.6%). The majority of the sample (n = 1036, 78.4%) were from Egypt and the remaining 21.6% of the sample (n = 286) were from the UAE. These demographic variables did not significantly differ between the two country's samples (see Table 1).

**Table 1. Results of descriptive analyses for all demographic variables (the total sample and for each country).**

|  | Total n = 1322 | Egypt n = 1036 | UAE n = 286 |
|---|---|---|---|
| Age M (SD) | 19.50 (1.54) | 19.45 (1.43) | 19.69 (1.88) |
| Gender (Female %) | 75.4 | 74.1 | 80.1 |
| Marital status % |  |  |  |
| Single | 90.6 | 91.0 | 89.2 |
| In a relationship | 7.0 | 6.6 | 8.4 |
| Married | 2.4 | 2.4 | 2.4 |
| Occupation % |  |  |  |
| Fulltime student | 96.4 | 96.7 | 95.1 |
| Part-time student and employed | 2.6 | 2.2 | 4.2 |
| Recently graduated and unemployed | 1.0 | 1.1 | 0.7 |

Note. M = mean; SD = standard deviation; UAE = United Arab Emirates.

**Table 2. Descriptive results and bivariate correlations between the principal study variables, stratified by country.**

|  | M (SD) | 2 | 3 |
|---|---|---|---|
| *Overall sample n = 1322* |  |  |  |
| 1. Depression | 7.22 (3.88) | .465** | -.113** |
| 2. Addictive SMU | 11.81 (5.07) | - | .007 |
| 3. Sense of control | 3.65 (1.36) | - | - |
| *Egypt n = 1036* |  |  |  |
| 1. Depression | 7.18 (3.91) | .447** | -.125** |
| 2. Addictive SMU | 11.86 (5.02) | - | -.003 |
| 3. Sense of control | 3.36 (1.67) | - | - |
| *UAE n = 286* |  |  |  |
| 1. Depression | 7.38 (3.79) | .534** | -.067 |
| 2. Addictive SMU | 11.61 (5.27) | - | .043 |
| 3. Sense of control | 3.63 (1.33) | - | - |

Note. M = mean, SD = standard deviation, UAE = United Arab Emirates, SMU = social media use,
**$p < .01$ (2-tailed).

The computation of bivariate correlations between the three principal variables revealed that depression was significantly and positively associated with addictive SMU (r = .47, p < .001) and negatively associated with sense of control (r = -.11, p < .001) but sense of control and addictive SMU were not significantly related (r = .007, p > .05). A similar pattern of results was evident when the data for each country was examined. Table 2 illustrates the results of the computed correlational matrices (total sample and one for each of the UAE and Egyptian samples).

## Mediation results

A mediation analysis was computed, and this revealed the following. The overall model was significant (F(2, 1319) = 182.526, $R^2$ = .217, p < .001). Path a was significant, depression was significantly associated with sense of control (β = -.04, SE = .01, 95%CI -.06, -.02) and so too was path b, sense of control was significantly associated with addictive SMU (β = .23, SE = .09, 95%CI .05, .40). Moreover, the total effect of depression on addictive SMU was also significant (β = .61, SE = .03, 95%CI .55, .67) and this association remained significant when the indirect effect of the mediator was examined. Sense of control emerged as a significant mediator of the association between depression and addictive SMU (path ab) (β = .62, SE = .03, 95%CI .56, .68). Fig 1 depicts the results of this mediation analysis and illustrates the total, direct and indirect effects.

## Discussion

This study sought to examine the potential mediational role of sense of control in the relationship between depressive symptoms and addictive SMU within the context of the COVID-19 pandemic. Doing so was highly pertinent given the evidence that rapidly emerged during the initial stages of the pandemic that rates of technology use and, in particular, excessive and problematic manifestations of SMU exponentially proliferated [49]. The results of this study revealed that depression was significantly and positively associated with addictive SMU (in confirmation of H1a). Sense of control was negatively associated with both depression (confirming H1b) and addictive SMU (confirming H1c). Moreover, sense of control significantly mediated the association of depression with addictive SMU (in confirmation of H2).

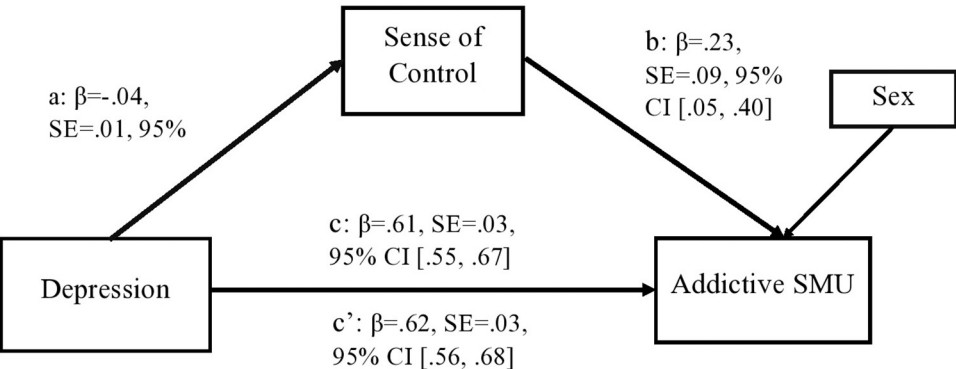

**Fig 1. Results of the examined mediation model.** Path a denotes the association between depression and sense of control; path b indicates the path between sense of control and addictive SMU; path c (the total effect) is reflective of the basic relationship between depression and addictive SMU; and path c' (the direct effect) denotes the link between depression and addictive SMU following inclusion of the mediator. SMU = social media use.

These results suggest that depressive symptoms may represent a risk factor for the development of addictive tendencies towards SMU. Our results produced statistically significant associations at the correlational level across all potential comparisons (i.e., the total sample and each of the country-specific subsamples) and resulted in statistically significant associations within the context of the mediation analysis. This provides additional substantiating evidence for the contentions of the I-PACE model and is in line with the findings of preceding empirical evidence [17, 18, 35, 50]. Specifically, where individuals experience depressive symptoms, this appears to elevate the degree of risk for the consequent onset of addictive SMU. Individuals may indeed actively seek out social media platforms and the access to information (about their experienced difficulties) that it facilitates and other individuals, in the online realm, can be readily and easily accessed from whom to elicit support and emotional comfort, thus increasing dependence and excessive use of the platform and device [17, 23].

Additionally, one of the potential mechanisms that creates the association between these variables appear to be the precipitous impact that depressive symptoms exact on individuals' sense of control. This suggestion appears sound when considered within the context of preceding research as well as the current understanding of depressive illness. Specifically, a multiplicity of research confirms that individuals with depression hold self-denigrating beliefs centred around their own lack of capacity to successfully manage the demands and responsibilities of their lives [51–53]. They frequently believe that their lives and their futures are out of their control and commensurately do not possess the capacity to execute some form of control [54, 55]. This is one of the principal contentions on which the cognitive model of depression is based [54]. Additionally, previous research also indicates that an elevated sense of control predicts lower levels of psychological distress, greater adaptation to stressful life events (e.g., transition to parenthood), and buffers the impact that multiple stressful life events have on the consequent development of mental health difficulties such as depression [38, 56, 57]. Thus, the evidenced association of depression and sense of control in this study appears to concur with preceding literature.

Our results also concur with existing research by highlighting a potential target for psychological intervention where individuals are assessed to be at risk for the onset of addictive SMU. Interventions that have specifically targeted the enhancement of individuals' sense of self-control and their self-efficacy beliefs have demonstrated efficacious outcomes in relation to depression, anxiety, stress, general wellbeing, improved resilience, health-related behaviours,

and greater pain tolerance [58–60]. Where individuals with depressive symptomology are assessed to be at risk for addictive SMU, a focused psychological intervention designed to target the elevation of their sense of control and self-efficacy may produce an analgesic effect and diminish depressive symptoms; however, this contention should be definitively tested within the context of a rigorous randomized controlled trial.

The results of the mediation analysis also suggest that improving individuals' sense of control may hypothetically mitigate the development of addictive SMU, however, as with the above suggestion in relation to depression, a definitive conclusion cannot be drawn given the cross-sectional design of the present study. The COVID-19 pandemic precipitated immense uncertainty. Many individuals struggled to manage the profound and relatively instantaneously imposed restrictions to daily life, the uncertainty of not knowing how the pandemic would proceed, and the feelings of loss, both tangible (i.e., loss of income, employment, death of loved ones) and abstract (i.e., loss control and certainty), that invariably accompanied the experience of the pandemic [1, 3]. Evidence that emerged during the pandemic demonstrated that some individuals who were unable to resiliently manage these burdensome aspects of the pandemic developed dysfunctional coping strategies such as addictive SMU [28, 37]. To escape the reality of a monumentally challenging life experience, some individuals with pre-existing mental health difficulties and/ or risk factors (e.g., a depressive illness) directed their attention towards the online realm to garner psychosocial support, or as a means of distraction, or, for some, this served a mood modification function [10]. Research indicates that individuals with higher levels of perceived control tend to be more resilient and are able to produce functional coping strategies when navigating distressing and uncertain experiences. Moreover, the experience of functional coping, successfully managing the demands of a challenging life experience, elicits positive emotions (i.e., wellbeing, joy, relief, mastery) and these, in turn, are likely to reduce the desire to turn one's attention away from the real-world towards online interactions. Therefore, this reinforces this study's proposition that the promotion of perceived sense of control in individuals at risk of mental health difficulties during stressful circumstances is a worthwhile psychotherapeutic course of action, and this is substantiated by our findings.

## Limitations and directions for future research

The following limitations should be borne in mind. First, the cross-sectional design of the study precludes a determination of potential causality. This is especially relevant when considering the principal contention of the examined mediational model. Despite evidence of the potential ameliorative effect that elevating an individual's sense of control may have on both diminishing depressive symptoms and reducing the likelihood that addictive SMU will ensue, a longitudinal design with multiple assessment points following the delivery of some form of intervention would be needed to reliably assess the impact that improving sense of control might have, if any, on these psychopathological outcomes. Potential interventions to be tested could take the form of meditation [61], those informed by the concept of salutogenesis [62], strengths-based interventions [63], or programs targeting self-defeating cognitions and beliefs [64]. Thus, while this study has identified a potential target for intervention (i.e., sense of control), the effect of such an intervention should be determined in future studies.

Second, only sense of control was assessed as a mediator. There may be any number of alternate constructs that might similarly mediate this relationship, constructs that were not assessed on this occasion. Specifically, the cacophony of factors that have been shown to be associated with or reflective of positive mental health may be expected to demonstrate a similar effect and, conversely, factors known to compound mental health outcomes may also demonstrate an association between these variables. For example, mindfulness, valued living,

committed action, personality, and lifestyle factors such as eating behaviour or the consumption of tobacco and/or alcohol might be considered [65]. Furthermore, constructs that have been shown to be associated with addictive SMU might also be considered for inclusion in further analyses–these might include internet gaming disorder [9], attachment style [46] or personality traits [66].

Third, participants' daily duration of social media use or the specific social media platforms that the sample preferred were not measured. These variables are highly relevant to any consideration of addictive SMU as duration of use, and the nature of social media use appears consistently related to the onset of addictive tendencies [4, 10]. Additionally, variations in the nature of SMU (i.e., whether use is active or passive) differentially impacts the development of comorbid psychopathological outcomes. Specifically, active engagement on social media platforms (e.g., posting status updates, writing comments to others, and uploading photos) appears more prevalently related to the development of addictive SMU whereas passive use (i.e., browsing content and reading others' comments) is associated with greater levels of envy, depression, and anxiety [7, 67]. Future studies should specifically assess duration of use, the features and platforms preferred, as well as whether SMU is active or passive to enable a determination of their differential effect.

Finally, the reliability and validity of the BSMAS in this population remains uncertain. The scale itself was originally developed with specific reference to Facebook use and while this scale has now been used extensively used, including in Arabic-speaking samples [4], its results should be interpreted with caution in the absence of an established psychometrically validated version.

## Conclusion

The result of this study indicates that, under environmental conditions of immense psychological distress–such as a global pandemic–individuals with pre-existing mental health difficulties, in particular depressive disorders, are at an especially elevated risk for the development of addictive SMU.

## Implications

This study suggests that targeting individuals' sense of control may diminish the potential risk for addictive SMU that is presented by the confluence of pre-existing mental health difficulties and environmental stress. It has long been noted that individuals with depression do indeed report a diminished sense of subjective control over their lives and the events that occur within it as well as the commensurate finding that, where individuals possess higher levels of control, the risk of associated psychological difficulties such as anger, anxiety or depression, precipitously diminishes [68, 69]. Given that this study now indicates that this relationship also occurred within the context of the COVID-19 pandemic and was elevated by the presence of co-occurring excessive SMU, it would be worthwhile exploring the potential utility of psychological interventions specifically targeting the elevation of sense of control amongst who present with the confluence of these two psychological difficulties. The principal implication of this study is therefore the identification of a potential target for psychotherapeutic intervention for individuals at elevated risk of depression and comorbid excessive SMU. As sense of control represents a form of maladaptive thinking [38], a cognitively-oriented approach seems most prudent. Positive psychology interventions have also proven to be effective in combating diminished control and elevating wellbeing [70, 71], and their potential utility within this context and for individuals with this specific form of presentation, should be explored. Additionally, where individuals find themselves in circumstances that are by their very definition

uncertain, such as a global pandemic, structured programs of psychoeducation by prove beneficial as the provision of context-specific information may create a degree of certainty.

## Supporting information

**S1 File. Complete dataset for the study (SPSS datafile).** Dataset on which the study's analyses are based.
(CSV)

## Author Contributions

**Conceptualization:** Zahir Vally.

**Data curation:** Zahir Vally.

**Formal analysis:** Zahir Vally.

**Methodology:** Zahir Vally, Mai Helmy, Louis Fourie.

**Writing – original draft:** Zahir Vally.

**Writing – review & editing:** Mai Helmy, Louis Fourie.

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
