## [Decision Letter · Decision Letter 0]

22 Feb 2023

PONE-D-23-02120Sense of control mediates the association between depression and addictive social media use during the COVID-19 pandemicPLOS ONE

Dear Dr. Vally,

Thank you for submitting your manuscript to PLOS ONE. After careful consideration, we feel that it has merit but does not fully meet PLOS ONE’s publication criteria as it currently stands. Therefore, we invite you to submit a revised version of the manuscript that addresses the points raised during the review process.

We look forward to receiving your revised manuscript.

Kind regards,

Sally Mohammed Farghaly

Academic Editor

PLOS ONE

4. Please ensure that you include a title page within your main document. You should list all authors and all affiliations as per our author instructions and clearly indicate the corresponding author.

Reviewers' comments:

Reviewer's Responses to Questions

**Comments to the Author**

1. Is the manuscript technically sound, and do the data support the conclusions?

Reviewer #1: Partly

Reviewer #2: Yes

2. Has the statistical analysis been performed appropriately and rigorously? 

Reviewer #1: Yes

Reviewer #2: Yes

3. Have the authors made all data underlying the findings in their manuscript fully available?

Reviewer #1: No

Reviewer #2: Yes

4. Is the manuscript presented in an intelligible fashion and written in standard English?

Reviewer #1: No

Reviewer #2: Yes

5. Review Comments to the Author

Reviewer #1: Dear authors,

It is a pleasure to review this manuscript. Thank you for giving such great opportunity.

In general, the topic is good and adds significant value, however, the manuscript need to be organized and summarized.

Please follow the following comments;

Title: I suggest to be modified as the following: sense of control mediating role on the relationship between depression and addictive social media use during COVID -19 pandemic

Abstract:

- Setting where data were collected should be added

- Research design also needs to be added

- Subjects must be stated clearly( you said participants ….which participants it must be delineated)

- Sampling technique need to be given also

- Statistical values must added in results that give impression about the relationship and the mediation effect

Introduction:

- Introduction need to be summarized and focused

- Many Paragraphs are too long especially in page 4

- Hypotheses should be organized and avoid using pronouns as we (you said we hypothesized)

Method:

This section should be carefully organized

- Design must be stated at the first

- Sampling technique must be mentioned

- Classes of students must be clearly stated

- Is the concept "federal" matched with Menoufia University in Egypt and UAE University?

- Setting not clear ( name of colleges where data were collected with brief description of settings required)

- What are the measures used to ensure validity and trustworthy of data collected from subjects? ……. Responses depend on self- reporting of subjects

- Is the 2-item scale used to measure sense of control is valid? – Is it able measure sense of control of students using 2 items?

Results:

- Well-presented however, I need to know why more than three quarters of students are females?

Discussion:

- Too long with many statements redundant with introduction

- It not follow guidelines of effective debate ( discussion need to be summarized using results of similar and contradictory studies)

- replace the pronouns we – our with the current study

Conclusion:

Conclusion of the study should be stated (I cannot find any section regarding conclusion).

Implications:

There is a must to add practical implications to the study (I cannot find any practical implications to the study).

Reviewer #2: Discussion and conclusion

Briefly comment on why the study is necessary without repeating the sentences used in the introduction, and include international studies. Kindly use this articles.

Summary of key findings (primary outcome measures, secondary outcome measures, results as they relate to a prior hypothesis); compare with findings from other studies.

6. PLOS authors have the option to publish the peer review history of their article (what does this mean?). If published, this will include your full peer review and any attached files.

Reviewer #1: No

Reviewer #2: No

---

## [Author Response · Author response to Decision Letter 0]

3 Apr 2023

Response to comments following review.

Editorial comments.

1. The paper has been revised according to the journal’s style guidelines – headings, file naming etc.

2. A deidentified dataset is now made available as a supplementary file.

3. The fact that written informed consent was obtained in clarified in the methods section.

4. A title page is included in the main document. 

Reviewer 1 comments.

1. We have modified the title as follows: we trust this change will find your positive approval. The association between depression and addictive social media use during the COVID-19 pandemic: The mediating role of sense of control.

2. The abstract has been expanded by adding the following details as you requested: setting, design, subjects, sampling technique, and statistical results. 

3. Introduction: 

• the Intro has been edited as far as possible with the view to shorten it. Paragraphs have also been abbreviated to ensure none are longer than 5 sentences at the most. To truncate this any further would result in a substantial loss of essential information. 

• In sum, the Introduction is laid out as follows: the mental health consequences of COVID are presented, including specifically the rise in addictive SMU; addictive SMU is then presented (description of features, associated mental health consequences, and with specific reference to COVID); sense of control is introduced and associated literature reviewed with the specific intention to introduce the proposition for its likely mediational properties; and finally, a theoretical framework. We regard this format as logical and the best approach in constructing the argument for the research question and its hypotheses.

• Any use of the ‘we’ pronoun has been removed. 

4. Method: 

• The following edits and/or additions have been made in concurrence with your requests: method section now commences with description of the study design, sampling technique is included, description of the targeted classes from students were drawn.

• The inherent limitation of selective reporting and social desirability when using self-report measures is acknowledged and its potential consequences discussed, at length, in the limitations section. This is not uncommon; indeed, it is the norm in psychological research. However, participants were provided with information in the participant info form, among others, of anonymity of data, protection of data, and who would have access to the data they reported. These measures were designed to assure participants and ensure accurate reporting of data, as far as this is possible within the confines of a cross-sectional study. 

• The 2-item scale has been extensively used in psychological research, including in relation to COVID research. It has been shown to be a valid measure of the construct. These studies are referenced in substantiation of its use.

5. Results: The gender distribution of the sample mirrors the gender distribution of the sampling population and is therefore a valid representation of the population. This is not the result of selective sampling. Moreover, gender has been included as covariate in the mediation model to account for any potential variation, if any. 

6. Discussion: This section has been edited and modified substantially with the view to truncate it as well as construct a thorough interrogation of the results. The only consideration here is that there are very few previous studies – barring one – that serve as a neat point of comparison for our results (thus enabling comparative discussion, of similarities and contradictions etc). In fact, the study’s research question – confirmation or disconfirm of the hypothesized mediation model is discussed in relation to preceding literature and the I-PACE model, which informed the research question. We believe this to be a prudent course of action in constructing the discussion.

7. Conclusion: paragraph written.

8. Implications: paragraph written. 

Reviewer 2 comments:

1. Discussion: This section has been edited and modified substantially with the view to truncate it as well as construct a thorough interrogation of the results. The only consideration here is that there are very few previous studies – barring one – that serve as a neat point of comparison for our results (thus enabling comparative discussion, of similarities and contradictions etc). In fact, the study’s research question – confirmation or disconfirm of the hypothesized mediation model is discussed in relation to preceding literature and the I-PACE model, which informed the research question. We believe this to be a prudent course of action in constructing the discussion.

2. Conclusion: paragraph written.

---

## [Decision Letter · Decision Letter 1]

15 Jun 2023

PONE-D-23-02120R1The association between depression and addictive social media use during the COVID-19 pandemic: The mediating role of sense of controlPLOS ONE

Dear Dr. Zahir,

Thank you for submitting your manuscript to PLOS ONE. After careful consideration, we feel that it has merit but does not fully meet PLOS ONE’s publication criteria as it currently stands. Therefore, we invite you to submit a revised version of the manuscript that addresses the points raised during the review process.

We look forward to receiving your revised manuscript.

Kind regards,

Sally Mohammed Farghaly

Academic Editor

PLOS ONE

Journal Requirements:

Reviewers' comments:

Reviewer's Responses to Questions

**Comments to the Author**

1. If the authors have adequately addressed your comments raised in a previous round of review and you feel that this manuscript is now acceptable for publication, you may indicate that here to bypass the “Comments to the Author” section, enter your conflict of interest statement in the “Confidential to Editor” section, and submit your "Accept" recommendation.

Reviewer #1: All comments have been addressed

Reviewer #3: All comments have been addressed

2. Is the manuscript technically sound, and do the data support the conclusions?

Reviewer #1: Yes

Reviewer #3: Yes

3. Has the statistical analysis been performed appropriately and rigorously? 

Reviewer #1: Yes

Reviewer #3: Yes

4. Have the authors made all data underlying the findings in their manuscript fully available?

Reviewer #1: No

Reviewer #3: Yes

5. Is the manuscript presented in an intelligible fashion and written in standard English?

Reviewer #1: Yes

Reviewer #3: Yes

6. Review Comments to the Author

Reviewer #1: Dear authors,

I am more than pleased for considering our comments in this revised manuscript.

In general, it is a good attempt that gives clear evidence to literatures.

There are some comments and questions that could strengthen your manuscript;

- How the researchers conducted snowball sampling?

- Heading "materials and methods" should be replaced with materials and method

- In discussion : replace "the potential mediational role of variation in sense of control" with the potential mediating role of sense of control

- Implications are still very weak- how this study will add to the clinicians

Reviewer #3: 1. In the part of Abstract, A combination of convenience and snowball sampling were used. However, in the part of Procedure and participants, his study employed a convenience sampling approach to collect data from college aged

participants.

2. The first sentence of the discussion should be followed by the second paragraph of the discussion.

3. There are no disscussion on the the relationship between depressive symptoms and SMU in the third paragraph of the discussion.

4. Why Table 1 and Table 2 were listed stratified by country, but mediation analyses were not stratified by country

7. PLOS authors have the option to publish the peer review history of their article (what does this mean?). If published, this will include your full peer review and any attached files.

Reviewer #1: No

Reviewer #3: No

---

## [Author Response · Author response to Decision Letter 1]

5 Aug 2023

Response to reviewer comments

Reviewer 1

Comment: How the researchers conducted snowball sampling?

Response: A sentence has been added to clarify how this was achieved on pages 8-9.

Comment: Heading "materials and methods" should be replaced with materials and method.

Response: Done.

Comment: In discussion : replace "the potential mediational role of variation in sense of control" with the potential mediating role of sense of control

Response: Done.

Comment: Implications are still very weak- how this study will add to the clinicians.

Response: This has been substantially expanded across pages 18-19.

Reviewer 3.

Comment: In the part of Abstract, A combination of convenience and snowball sampling were used. However, in the part of Procedure and participants, his study employed a convenience sampling approach to collect data from college aged

participants.

Response: the text in the Method section has been amended to include this omission.

Comment: The first sentence of the discussion should be followed by the second paragraph of the discussion.

Response: Done.

Comment: There are no discussion on the relationship between depressive symptoms and SMU in the third paragraph of the discussion.

Response: Paragraph 2 on page 14 has been expanded substantially to expand on the discussion related to the association of these variables.

Comment: Why Table 1 and Table 2 were listed stratified by country, but mediation analyses were not stratified by country

Response: Country-specific descriptive data is provided for descriptive purposes only. Mediation analyses were conducted separately as the UAE sample would be too small on which to run a sufficiently powered analysis. Thus, a mediation analysis on the total sample was deemed to be the most prudent approach, while the descriptive data relating to each country may be of interest to some readers.

---

## [Editor Report · Decision Letter 2]

21 Aug 2023

The association between depression and addictive social media use during the COVID-19 pandemic: The mediating role of sense of control

PONE-D-23-02120R2

Dear Dr. Zahir,

We’re pleased to inform you that your manuscript has been judged scientifically suitable for publication and will be formally accepted for publication once it meets all outstanding technical requirements.

Kind regards,

Sally Mohammed Farghaly

Academic Editor

PLOS ONE
---

## [Editor Report · Acceptance letter]

1 Sep 2023

PONE-D-23-02120R2 

The association between depression and addictive social media use during the COVID-19 pandemic: The mediating role of sense of control 

Dear Dr. Vally:

I'm pleased to inform you that your manuscript has been deemed suitable for publication in PLOS ONE. Congratulations! Your manuscript is now with our production department. 

Kind regards, 

on behalf of

Professor Sally Mohammed Farghaly 

Academic Editor

PLOS ONE